# Modular quantum computation in a trapped ion system

Kuan Zhang[1,2]*, Jayne Thompson[3]*, Xiang Zhang[1,4], Yangchao Shen[1], Yao Lu[1], Shuaining Zhang[1], Jiajun Ma[1,5], Vlatko Vedral[1,3,5,6], Mile Gu[3,7,8]* & Kihwan Kim [1]*

Modern computation relies crucially on modular architectures, breaking a complex algorithm into self-contained subroutines. A client can then call upon a remote server to implement parts of the computation independently via an application programming interface (API). Present APIs relay only classical information. Here we implement a quantum API that enables a client to estimate the absolute value of the trace of a server-provided unitary operation $U$. We demonstrate that the algorithm functions correctly irrespective of what unitary $U$ the server implements or how the server specifically realizes $U$. Our experiment involves pioneering techniques to coherently swap qubits encoded within the motional states of a trapped $^{171}\text{Yb}^+$ ion, controlled on its hyperfine state. This constitutes the first demonstration of modular computation in the quantum regime, providing a step towards scalable, parallelization of quantum computation.

[1] Center for Quantum Information, Institute for Interdisciplinary Information Sciences, Tsinghua University, 100084 Beijing, China. [2] MOE Key Laboratory of Fundamental Physical Quantities Measurement & Hubei Key Laboratory of Gravitation and Quantum Physics, PGMF and School of Physics, Huazhong University of Science and Technology, 430074 Wuhan, China. [3] Centre for Quantum Technologies, National University of Singapore, Singapore 117543, Singapore. [4] Department of Physics, Renmin University of China, 100872 Beijing, China. [5] Department of Atomic and Laser Physics, Clarendon Laboratory, University of Oxford, Oxford OX1 3PU, UK. [6] Department of Physics, National University of Singapore, Singapore 117551, Singapore. [7] School of Mathematical and Physical Sciences, Nanyang Technological University, Singapore 637371, Singapore. [8] Complexity Institute, Nanyang Technological University, Singapore 637335, Singapore. *email: zhangkuan13@gmail.com; thompson.jayne2@gmail.com; gumile@ntu.edu.sg; kimkihwan@mail.tsinghua.edu.cn

Whan Google upgrades their hardware, applications that make use of Google services continue to function without needing to update. This modular architecture allows a client, Alice, to leverage computations done by a third party, Bob, without knowing any details regarding how these computations were executed. Modularity is enabled by an interface—an established set of rules that specify how Alice delivers input to Bob, and how Bob returns relevant output to Alice. Once agreed, Alice can design technology that makes use of the Bob's service as subroutines, while remaining blissfully ignorant of their implementation. Known as APIs (application programming interfaces), such interfaces are now industry standard. Their adoption is almost universal—from specifying how we leverage pre-built software packages as subroutines to how we interface remotely with present-day quantum computers.

Present interfaces assume only classical information is exchanged, limiting the scope of collaborative quantum computing. What happens when this information exchange is allowed to be quantum? Consider the scenario where Bob offers a service to implement some unitary operation $U$. A client, Alice wishes to evaluate the normalized trace $T(U) = \mathrm{tr}(U)/2^n$ by calling on Bob's service as a subroutine. If this can be achieved, the benefits are two-fold. Alice can treat Bob's service as a black-box. She need not know anything about the quantum circuits that synthesize $U$. In addition, Alice can use the same device to evaluate the normalized trace of a different unitary $U'$, by exchanging Bob's service for another.

This is, in fact, impossible. To see this, note that $T(U)$ depends on the global phase of $U$—a quantity that is unphysical. Therefore, its determination would enable Alice to measure an unphysical quantity. Thus, the standard quantum algorithm for estimating $T(U)$, known as DQC1[1], cannot operate by offloading synthesis of $U$ to a third party (see Fig. 1a). Indeed, the design of devices that realize complex $U$-dependent processes, given some unknown $U$, has received considerable attention[2–10]. In this context, several no-go results have been established[11–15], motivating recent works in identifying what sacrifices or restrictions are necessary to circumvent these no-go theorems[14–20].

Here, we report on the experimental implementation of a workaround for the DQC1 algorithm. The key observation is that while $T(U)$ depends on the global phase, its modulus does not. The resulting protocol—modular DQC1—enables us to evaluate $|T(U)|$ by outsourcing implementation of $U$ to a third party[15]. We successfully use it to evaluate $|T(U)|$ for 19 different unitary operations. The quantum circuit for the client remains the same for each $U$—guaranteeing true modular architecture. The physical implementation involves a new implementation of the controlled swap (CSWAP) gate—coherently swap two motional modes of an ion trapped in a three-dimensional harmonic oscillator, controlled on the internal levels of the trapped ion. Our experimental techniques are scalable, resilient to noise on part of the client, and chaining multiple iterations enables a modular variant of Shor's factoring algorithm that requires fewer entangling gates[21]. This presents the first demonstration of a modular quantum algorithm and provides an important step towards collaborative quantum computing.

## Results

**Framework**. The modular DQC1 algorithm can be understood by dividing its actions into two separate parties, which we refer to here as server and client. The server, Bob, offers the service of implementing an $n$-qubit unitary process. Interaction with a client, Alice, is enabled by a publicly announced quantum interface. The interface specifies a designated Hilbert space of a designated quantum system $S$ in which client and server are to exchange quantum information. For simplicity, we assume that the agreed system and Hilbert space used by client to send input quantum states to the server in the same as that used by the server to deliver output quantum states to the client. In principle, this need not be the case (see methods for formal definition). Bob is not constrained to preserve information stored in any other degrees of freedom within $S$. This is an important point. If Alice is guaranteed that Bob will preserve certain additional degrees of freedom, she is able to synthesizes certain $U$-dependent process that would otherwise be impossible[14,15]. Our goal is to take on the

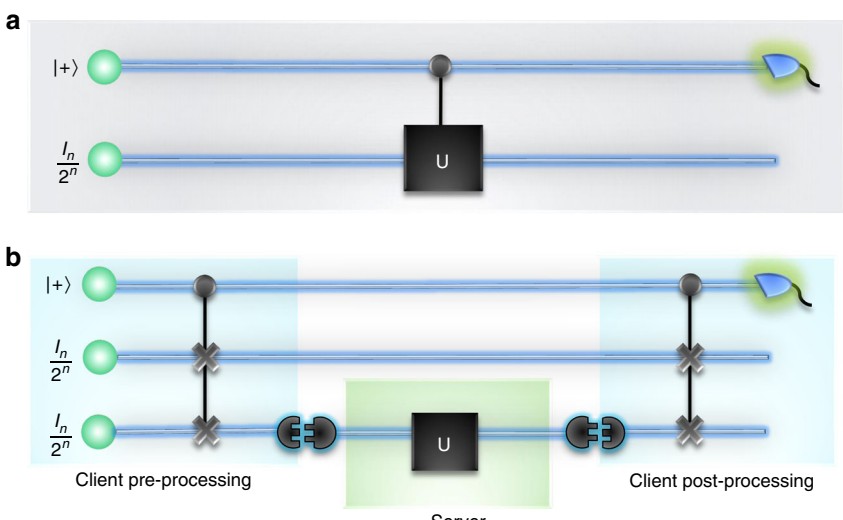

**Fig. 1** The DQC1 and modular DQC1 algorithms. **a** The standard DQC1 algorithm operates by applying $U$ on an $n$-qubit register controlled by a pure qubit initialized in state $|+\rangle$. $T(U)$ can then be estimated through appropriate measurements on the control qubit. This algorithm cannot leverage a third party to implement $U$ as it is impossible to add a control to an unknown unitary[11]. **b** The modular DQC1 algorithm evaluates $|T(U)|$ in a way in which $U$ can be out-sourced to a third party. Here, Alice introduces a second $n$-qubit register. She then sends the server one of the $n$-qubit registers via a specified interface (this could be the original register, or involve first mapping the register into a medium suitable for communication via a SWAP gate). On the proviso that the server applies $U$ and return the result via the specified interface, Alice is able to estimate $|T(U)|$ by performing a $\sigma_1$ measurement on the control qubit

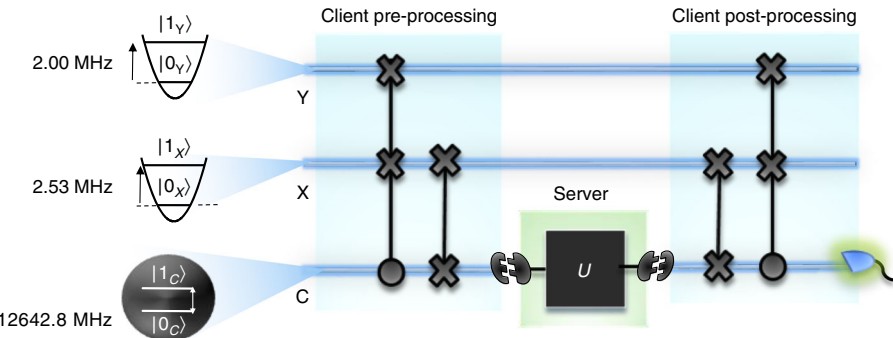

**Fig. 2** Modular DQC1 on a trapped ion. The modular DQC1 algorithm redesigned to function on a $^{171}$Yb$^+$ ion. Here, the control qubit C is encoded within two hyperfine levels of the S$_{1/2}$ manifold in the ion. Denote these by $|0_C\rangle = |F = 0, m_F = 0\rangle$ and $|1_C\rangle = |F = 1, m_F = 0\rangle$, where $F$ is the quantum number of total internal angular momentum and $m_F$ is the magnetic quantum number. The transition frequency between $|0_C\rangle$ and $|1_C\rangle$ is 12642.826 MHz. Qubits X and Y are encoded within the ground and first excited states of two radial motional modes in $^{171}$Yb$^+$, denoted as $|0_X\rangle$, $|1_X\rangle$ and $|0_Y\rangle$, $|1_Y\rangle$. The trap frequencies of modes X and Y are given by 2.53 and 2.00 MHz. After suitable preprocessing, information encoded within the control qubit can be forwarded to an external server via a suitable interface where the action of $U$ is out-sourced

role of Alice, and build a device that employs Bob's service as a subroutine to evaluate $|T(U)|$.

To do this, Alice begins with a bipartite system, consisting of $S$ to be delivered to Bob and some $A$ that she retains for the duration of the protocol. The protocol then contains two distinct tasks (see Fig. 1b):

Preprocessing—representing Alice's necessary actions of preparing some $\rho_1$ on the joint system $A \otimes S$ before delivery of $S$ to Bob;

Postprocessing—representing Alice's actions to retrieve $|T(U)|$ from the state $\rho_2 = U\rho_1 U^\dagger$ after receiving Bob's output. Here, $U$ represents the unitary process on $S$ implemented by Bob.

Alice can achieve this by taking a single pure qubit initialized in state $|+\rangle = (|0\rangle + |1\rangle)/\sqrt{2}$, together with two maximally mixed $n$-qubit registers. In the preprocessing stage, she coherently swaps the two registers, controlled on the pure qubit to obtain $\rho_1$. Alice then forwards one of the registers to Bob via the specified interface and awaits the result of Bob's computation. Upon receipt of this result, Alice enters the postprocessing stage. This involves a second application of the control swap gate. Measurement of the ancilla in the $\sigma_1 = |0\rangle\langle 1| + |1\rangle\langle 0|$ basis then has expectation value of $|T(U)|^2$, enabling efficient estimation of $|T(U)|$. Further details are shown in Fig. 1b.

The combination of preprocessing and postprocessing constitutes the modular DQC1 protocol. Critically, neither procedure depends on the physical means that Bob chooses to realize $U$. For instance, Bob could initially implement $U$ by applying physical operations directly on the system $S$. Alternatively, Bob could map the received quantum state to a more efficient physical platform for information processing, and implement $U$ on that platform. Alice's modular DQC1 protocol would function regardless. Moreover, Alice's preprocessing and postprocessing procedures are independent of matrix elements of $U$. This becomes pertinent in cases where $U$ could represent some unknown environmental process. The protocol then functions as a probe, able to efficiently estimate $|T(U)|^2$ for any such process without the need for full tomography.

**Implementation**. We demonstrate a proof of principle realization of modular DQC1 using a trapped $^{171}$Yb$^+$ ion in a harmonic potential when $n = 1$. In this special case, the protocol involves a system of three qubits. Qubit C represents the control, which is encoded into the internal states of $^{171}$Yb$^+$. Two registers, denoted as qubits X and Y, are encoded into the external motional levels of

$^{171}$Yb$^+$. Unitary operations on qubit C are performed by applying resonant microwaves[22,23]. Meanwhile entangling gates between the ancilla and the two registers are realized by applying counter-propagating Raman laser beams with appropriate frequency differences and phases[24–28].

The theoretical circuit for modular DQC1 has also been further tailored for the ion trap system. Notably, during both preprocessing and postprocessing, Alice has inserted an extra SWAP gate between qubit C and qubit X. The actions of these SWAP gates have no effect on the algorithms output, but benefit this particular set-up, as the control qubit in the ion trap system is most directly accessible—and thus the most practical one for outsourcing operations to a third party. For the proof-of-principle experiment, we simulate the scenario where Bob operates on C directly—with understanding that in more realistic scenarios, information within X is likely first mapped to some flying qubit to be delivered to Bob. Fig. 2 illustrates further details.

During preprocessing, the standard circuit design for synthesizing $\rho_1$ involves application of a CSWAP gate on registers X and Y with qubit C as the control. Since $\rho_1$ is input-independent, any means of preparing this state is equally valid. Here, we perform preprocessing without explicitly using the CSWAP gate, with no impact on practical usages and scalability (see Supplementary Notes 1 and 2). Subsequently, a second SWAP is then used to interchange information between X and C—enabling C to be used as the interface qubit.

During postprocessing, a second CSWAP gate is necessary for extracting information about $|T(U)|$ from $\rho_2$. This $\rho_2$ is input-dependent, thus the CSWAP gate needs to be synthesized online. In our experiment, we pioneer a technique to achieve this involving motional qubits and a sequence of Raman laser beams together with microwaves (see Fig. 3. The full implementation of CSWAP is illustrated in Supplementary Note 1). Appropriately configured measurements on qubit C via fluorescence detection will then have measurement outcomes with an expectation value of $|T(U)|^2$. Repeated applications of the protocol thus efficiently estimate $|T(U)|$ to any specified accuracy.

To characterize the faithfulness of our CSWAP operation, we find its $8 \times 8$ truth table. The implementation achieves a fidelity (classical gate fidelity[29]) of $0.85 \pm 0.02$ (see Supplementary Note 3 for details). In methods, we illustrate that effects of these imperfections can be mitigated—such that use of our CSWAP operations does not impact Alice's capability to efficiently estimate $|T(U)|$ to any fixed error.

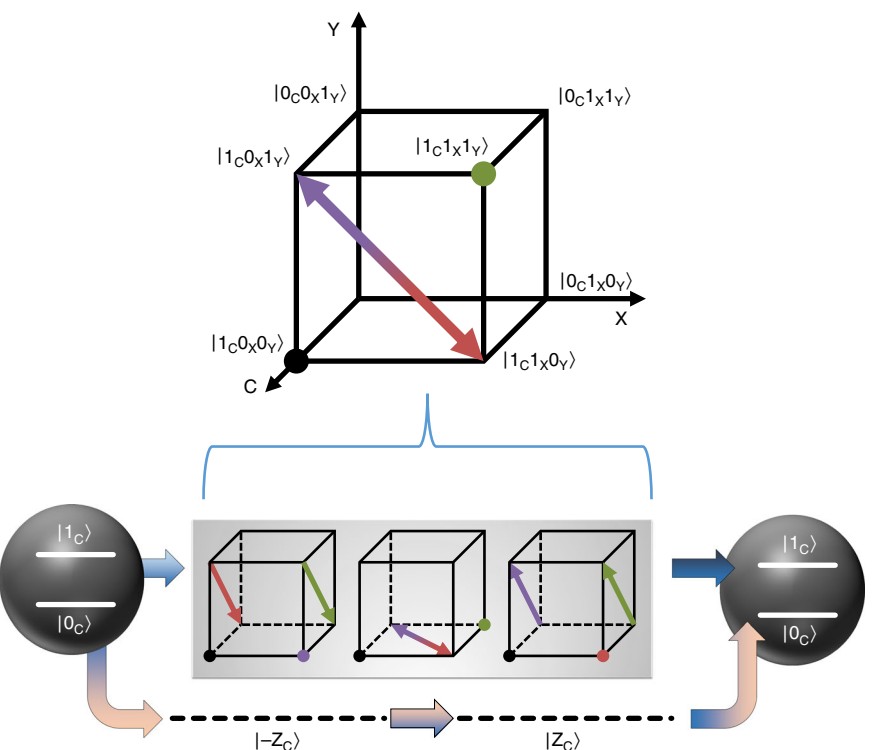

**Fig. 3** Implementation of the control SWAP gate. A CSWAP operation on X and Y represents coherently interchanging the populations of $|1_C 0_X 1_Y\rangle$ and $|1_C 1_X 0_Y\rangle$. In experiment this is achieved as follows: first, we temporarily shelve $|0_C\rangle$ into an ancillary Zeeman level $|\pm Z_C\rangle = |F = 1, m_F = \pm 1\rangle$ by microwave pulses. The two Zeeman levels $|Z_C\rangle$ and $|-Z_C\rangle$ are employed sequentially with equal duration, so that the AC stark shift and energy level jittering of both Zeeman levels cancel. The transition between $|0_C\rangle$ and $|\pm Z_C\rangle = |F = 1, m_F = \pm 1\rangle$ is realized by a microwave pulse with frequency $12642.819 \pm 9.507 m_F$ MHz. Meanwhile the SWAP operation that interchanges $|1_C 0_X 1_Y\rangle$ and $|1_C 1_X 0_Y\rangle$ is realized by three sequential Raman pulses (see Supplementary Note 1 and 2). Each Raman process is represented by a cube in the figure, where an arrow indicates that population is transferred, and a dot shows population is not transferred. Subsequently, the shelved $|0_C\rangle$ is restored by a second microwave pulse

**Experimental benchmarks**. To benchmark Alice's protocol, our experiment also needs to simulate the actions of the server Bob. Critically, we ensure that our experimental procedure for enacting Alice's modular DQC1 protocol in both preprocessing and postprocessing remains invariant regardless of which $U$ is implemented. Operationally, this enables us to simulate the following scenario:

1. Alice performs relevant preprocessing and walks away from her lab.
2. Bob then implements $U$ on C in her absence.
3. Alice can then return to perform estimation of $|T(U)|$ without specific knowledge, which $U$ was implemented, or what methodology Bob used.

This enables us to illustrate the core tenet of modularity—that the client's circuit does not need to change depending on $U$.

During benchmarking, we assess Alice's performance for a wide range of unitaries $U$. Specifically, these include unitaries of the form $U_\sigma(\chi) = \exp(-i\chi\sigma/2)$, where $\sigma \in \{\sigma_1, \sigma_2, \sigma_3\}$ involves all three possible Pauli operators, and $\chi \in \{0, \pi/6, \pi/3, \pi/2, 2\pi/3, 5\pi/6, \pi\}$ as shown in Fig. 4. For each choice of $U$, the protocol was executed 1000 times to obtain an estimation of $|T(U)|^2$—denoted as $M(U)$—with standard error of $\sim 0.02$.

We compare these experimental results to their theoretical predictions in Fig. 4a. As we can see, the experimental estimations of $|T(U)|^2$ are significantly lower than their true values. Fortunately, Alice is able to calibrate her device to account for these errors. To do this, she assumes the resulting estimations $M$ are offset by a scaling factor of $\lambda$, i.e., $M(U) = \lambda |T(U)|^2$. Alice can determine $\lambda$ by first benchmarking her device against an 'identity server' (e.g., preforming preprocessing and postprocessing without calling on the services of Bob). The effectively evaluates $M(I)$—which should output 1 under ideal conditions, and thus enables immediate estimation of $\lambda$. She can then scale all results by a factor of $1/\lambda$. As this scaling is independent of how the server implements $U$, this form of error correction does not impact modularity of the procedure.

In our experiment, the value $\lambda$ is determined to be $0.69 \pm 0.02$ to a confidence level of 95%. The re-calibrated estimations for $|T(U)|^2$ are plotted with theoretical predictions in Fig. 4b. As we can see, Alice's estimations of $|T(U)|^2$ are now in good agreement with their true values. As such, we illustrates that modular DQC1 can continue to operate in today's experimental conditions.

## Discussion

Here, we experimentally demonstrated the first modular quantum protocol—a variation of the standard DQC1 protocol that allows a device to determine the normalized trace of a completely unknown unitary process $U$. The experiment illustrates how Alice can outsource part of the computation to Bob—namely the realization of $U$. Alice needs no knowledge of how Bob chooses to realize $U$. The only information Alice and Bob need to share is an agreement on how to communicate quantum information to each other. Modular architecture has been critical in distributed classical computing. Our experiment presents its analog in the quantum regime.

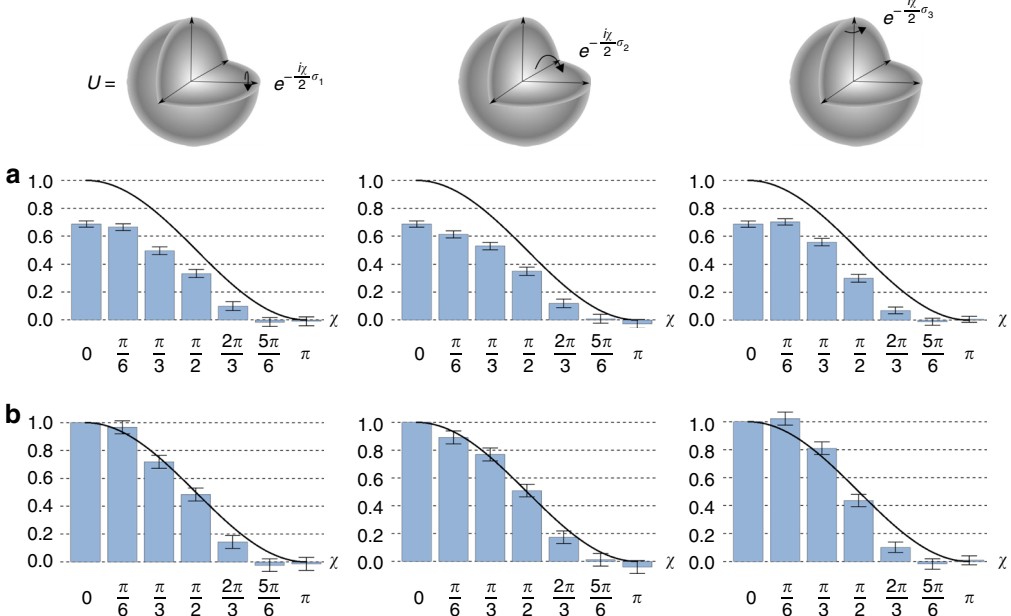

**Fig. 4** Experimental results. Benchmarking results for modular DQC1 with 19 different server supplied unitary operations $U_{k,\chi} = \exp(-i\chi\sigma_k/2)$, where $\chi$ ranges over $\{0, \pi/6, \pi/3, \pi/2, 2\pi/3, 5\pi/6, \pi\}$ and $\sigma_k = \sigma_1, \sigma_2, \sigma_3$ ranges over all three standard Pauli directions. **a** displays resulting experimental estimations of $T(U_{k,\chi})$ (blue bars), as compared to theoretic predictions (black lines). The disparity is due to decoherence. **b** Calibrating to account for this decoherence enables agreement between theory and experiment. Error bars in both **a** and **b** means a confidential interval with 95% confidence

Our implementation involved the design and realization a coherent quantum controlled swap gate, swapping two motional modes of a trapped ion depending on its internal hyperfine states. This technique presents a more favorable means of scaling than encoding qubits only within the internal states of ions. In Supplementary Note 1, we illustrate that our techniques can be adapted to efficiently swap two registers containing many motional modes, controlled on the hyperfine states of single ion. Meanwhile employing higher-energy excitations of the motional modes can enable potential coherent swaps of continuous variable degrees of freedom. These techniques provide possible means of realizing a number of interesting quantum protocols, including quantum anomaly detection[30], and quantum computing with continuous variable encodings[31].

## Methods

**Formal framework**. A quantum application programming interface (API) specifies a public agreement between a client and server in how to communicate quantum information[15]. In particular, an interface $\mathcal{I}$ involves two tuples:

1. $\mathcal{I}_{\text{in}} = (S_{\text{in}}, \mathcal{H}_{\text{in}}, \mathcal{B}_{\text{in}})$ consisting of the physical system $S_{\text{in}}$, and precise Hilbert space $\mathcal{H}_{\text{in}}$ that Alice promises to use to deliver information to the server, as well as the computational basis $\mathcal{B}_{\text{in}}$, which Alice will use to encode this information.
2. $\mathcal{I}_{\text{out}} = (S_{\text{out}}, \mathcal{H}_{\text{out}}, \mathcal{B}_{\text{out}})$ consisting of the exact physical system $S_{\text{out}}$, Hilbert space $\mathcal{H}_{\text{out}}$, and computational basis $\mathcal{B}_{\text{out}}$, which the server will use to return output quantum information to Alice.

We then say that a server, Bob, implements $U$ via interface $\mathcal{I}$ if on delivery of $|\phi\rangle$ encoded within $\mathcal{I}_{\text{in}}$, Bob will return $U|\phi\rangle$ encoded within $\mathcal{I}_{\text{out}}$. Note that in many settings, our experiment included, $\mathcal{I}_{\text{in}} = \mathcal{I}_{\text{out}}$.

Once an interface is agreed. Alice can then design modular algorithms that take advantage of Bob's service. Formally, we define two possible classes of elementary actions

1. Implement some elementary circuit elements (e.g., a elementary quantum gate, a single-qubit measurement)
2. Call upon the server to act on $S_{\text{in}}$ and wait for reception of $S_{\text{out}}$

A modular quantum algorithm is then defined as a $U$-independent sequence of elementary actions that enable Alice to realize a quantum process $\mathcal{P}[U]$ whenever Bob implements $U$. In our experiment, $\mathcal{P}[U]$ was a quantum process whose output

allowed efficient estimation of $\text{tr}(U)$. The key advantages of this modular architecture is that it ensures

- Independence of realization—Alice's algorithm realizes $\mathcal{P}[U]$, irrespective of what sequence of physical operations Bob uses to implement $U$.
- Independence of function—If Alice wishes to realizes $\mathcal{P}[V]$, she does not need to modify her algorithm. She just needs to find a server that implements $V$ instead of $U$ via interface $\mathcal{I}$.

We note also that while in many practical scenarios, client and server would be spatially separated, this need not be the case. An examples of local APIs in the classical setting are software packages, where certain functions can be invoked as subroutines without needing to know their details.

**Client error calibration**. Here, we illustrate details of how Alice can calibrate her device to account for experimental noise in her set-up. Specifically, the expected output state of the circuit immediately prior to the measurement is

$$\rho_{\text{ideal}} = \frac{1}{2^{2n+1}} \begin{pmatrix} I^{\otimes 2n} & U \otimes U^{\dagger} \\ U^{\dagger} \otimes U & I^{\otimes 2n} \end{pmatrix}. \qquad (1)$$

Measurement in Pauli-X basis then yields the desired expectation value of $\langle\sigma_1\rangle_{\text{ideal}} = |T(U)|^2$. By the central limit theorem, she can thus estimate $|T(U)|^2$ to any specified accuracy $\epsilon$ by repeating the procedure $O(1/\epsilon^2)$ times.

In our actual experiment, Alice's device is not ideal. The dominant noise occurs during the implementation of CSWAP gate, caused by fluctuations in the magnetic field, trap frequencies, polarization, and intensity of the Raman lasers. This introduces decoherence, such that Alice obtains

$$\rho_{\text{exp}} = \lambda\rho_{\text{ideal}} + (1 - \lambda)\frac{I}{2^{2n+1}} \qquad (2)$$

in place of $\rho_{\text{ideal}}$, where $0 \leq 1 - \lambda \leq 1$ benchmarks the level of effective decoherence. Subsequent Pauli-X yields expectation values $\langle\sigma_1\rangle_{\text{exp}} = \lambda|T(U)|^2$. Alice can estimate the value of $\lambda$ by effectively running modular DQC1 using $U = I$, without making use of a third party service. Once $\lambda$ is determined, Alice can mitigate the effects of noise by setting her estimation to be $|T(U)|^2_{\text{est}} = \langle\sigma_1\rangle_{\text{exp}}/\lambda$, enabling an estimation of accuracy $\epsilon$ with $O(\lambda^{-2}\epsilon^{-2})$ server calls. Therefore, our modular DQC1 algorithm is resilient to experimentally dominant sources of noise on the part of client.

We note that in this entire procedure, Alice's actions does not depend on which unitary Bob implements, or how he chooses to implement this unitary. Thus, the noise-corrections do not affect the modular nature of the algorithm.

## Data availability
The data that support the findings of this study are available from the corresponding author upon reasonable request.

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

## Acknowledgements
This work was supported by the National Key Research and Development Program of China under Grants No. 2016YFA0301900, No. 2016YFA0301901, and the National Natural Science Foundation of China Grants No. 11574002, the National Research Foundation of Singapore (Fellowship NRF-NRFF2016-02), the John Templeton Foundation (grant 54914), and the National Research Foundation and the Agence Nationale de la Recherche joint Project No. NRF2017-NRFANR004 VanQuTe, the FQXi Large Grant: The role of quantum effects in simplifying adaptive agents and Huawei Technologies.

## Author contributions
K.Z., X.Z, Y.S., Y.L., and S.Z. developed the experimental system. J.T., J.M., V.V., and M.G. proposed the protocol. K.Z. implemented the protocol and led the data taking. K.K. supervised the experiment. K.Z., J.T., M.G., and K.K. wrote the manuscript.

## Competing interests
The authors declare no competing interests.
