## [Peer Review File · Nature Communications]

Reviewers' Comments:

Reviewer #1:

Remarks to the Author:

The authors examine a protocol for determining the $\text{Tr}|U|$. The experiment involves a single trapped ion and two motional modes. The conceptual idea is that U is performed on a second system controlled by Bob and Alice is allowed send and receive quantum information to and from Bob's system. The authors make an analogy to an API. The paper will be of interest to Nature Communications readers.

The experiment is interesting and includes a number of gate compilation methods to simplify the procedure. The authors clearly explain in the paper and the supplementary materials how the experiment is performed. The data is well presented and the error bars are explained. The error with decoherence is quite large and points to the limits of this experiment.

The weakness of the paper is the API motivation. It seems unnecessary and then the readers expectations are deflated by the procedure occurring in a single system. The authors also claim in the discussion that the modes of motion provide a favorable path to scaling over the internal degrees of freedom but this is poorly motivated.

Specific comments:

In the supplementary material, Figure 2d has a typo in the first column of the fourth row. Both states in the bottom register should be 1.

Reviewer #2:

Remarks to the Author:

In the manuscript "Modular Quantum Computation in a Trapped Ion System" the authors describe an experimental demonstration of the modular quantum computation protocol detailed in [15] using a trapped Yb ion. The procedure is a variant of the DQC1-procedure described in [2] which allows a client to evaluate the modulus of the trace of a unitary. The unitary is provided as a service by a server and applied to a qubit provided by the client via a quantum-API. The modularity stems from the fact that the client can be blind to the particulars of the implementation of the unitary provided by the server. This is a highly relevant topic in quantum computing since all current realizations show only classical modularity, i.e. different circuit elements can be rearranged, or ion-qubits moved. The paper is well written and clearly structured, with very neat illustrations. Some of the implementation only becomes understandable when studying the supplementary material. I think the result is highly relevant to the field of quantum information processing with trapped ions and of interest to anyone working in implementations of quantum algorithms and quantum computing hardware and architecture. I am not convinced it warrants the interest of the general natural science community for the following reason.

To me, the most impressive achievement is that the authors perform what is essentially a 3-qubit algorithm in a single-ion experiment by clever use of the different degrees of freedom provided by the system, namely the two qubit states, plus two auxiliary electronic Zeeman states, and 4 different motional states in the two radial modes. This shows how efficiently quantum information can be encoded in a trapped ion system. It also nicely illustrates the idea that in a server-client relationship, the implementation of a subroutine might be executed on a different kind of qubit platform with the exchange being handled by the API.

This approach also appears to me to contain a weak point of the paper. Since there is only one real qubit, i.e. quantum system with complete gate set, the unitary has to be executed on that one qubit. So the information Alice holds on her control qubit is stored temporarily elsewhere, and the qubit the server receives is copied back onto Alice's qubit, so that the same control field (uWave) that she used to prepare her state performs the unitary. This seems to me to go against the spirit of modularity the authors themselves have evoked since the operations of the server are not independent and arbitrary, they are in fact the same as the client's. This removes the main complication from a quantum API, namely how to achieve the interface between the qubit-sender and the qubit-receiver to ensure continuous coherent control. Since the two controls are identical here, the problem reduces to that of a regular algorithm with operations performed in a single module. While it makes the implementation work (which is, I reiterate, very impressive), I worry that it also weakens the point the authors are trying to make with the demonstration - modularity (i.e. actual modularity rather than modularity in principle).

I have enjoyed reading this paper and believe it demonstrates an interesting quantum algorithm that touches on a subject not previously experimentally investigated, but for the reason stated, a more quantum computer focused journal like Nature-QI might be a better venue for this work.

Smaller points:

p.2 "The physical implementation involves pioneering techniques to coherently swap two motional modes of an ion trapped in a 3D harmonic oscillator, controlled on the internal levels of the trapped ion."

While it is true that this particular implementation of a C-Swap gate is new, storing information temporarily in the motional modes to create quantum gates is an old idea, as this very process lies at the heart of the Cirac-Zoller gate. I feel "pioneering" is too strong a term.

- p.2 "(e.g. coherence in the photon number degree of freedom)" - at this point in the paper, the experimental system has not been introduced, which photon degrees of freedom are meant is not clear

- supplementary materials sections are labelled differently than the references to them in the manuscript (i.e. where is Supplementary II.C, referenced on p.3 bottom left?)

- figure 5: why are the experimental errors making all trace estimates systematically lower, rather than e.g. pushing them to 0.5 or some other value?

- fig 5 b and accompanying text: Rescaling of the experimental values works here because the error of implementing a single-qubit unitary is small and the errors come from the operations associated with the pre- and postprocessing operations. I imagine it will be much harder when the server uses an unknown number of gates to make some unknown U. Does this not mean the correction will become server- and server-implementation dependent which seems to go against the API philosophy (i.e. "swapping one server for another") that the paper is based on?

- in the supplementary, the preparation of the mixed state is not completely clear to me: It is created by a running different circuits, one for each of the 4 basis states and averaging the results. Are 1000 repetitions performed for each circuit and the result averaged, or 250 for each giving 1000 in total? Also, if this is part of a larger quantum program with multiple API calls, running it in sequence for each combination might not be practical. Is there a way to create a mixed state resource online (perhaps by using another motional mode that is traced out?)

minor technical comments:

there are some language errors in the supplementary:

examples: p2: "for THE general n case," "requires the implementation of THE following unitary," p3: "requires A single sigma measurement"

main text: figure 4 caption "is A consequence"

figure 3: the bottom figure seems to be missing some colored circles, I assume the 3-cube sequence is meant to show the purple and red dot changing places while leaving the black and green dot unmoved by the end.

stylistic comments (these are a matter of preference):

The mention of companies Google and IBM in the first paragraph tries to emulate the introduction to [1] but doesn't do it well. The IBM quantum computer would perform worse on the circuit the authors have implemented here. Generic examples without commercial brands might be better.

p.2 "no other promises are made." - too strong, a common phase reference is also necessary to make it work

清华大学交叉信息研究院

Tsinghua University Institute for Interdisciplinary Information Sciences

北京市海淀区清华大学 FIT 楼, 邮编 100084, 电话: (010)62781693, 传真: (010)62797331

Tel: +86-10- 62781693

Fax: +86-10-62797331

Website: <http://iis.tsinghua.edu.cn>

Reply to Reviewer #1

"The authors examine a protocol for determining the $\text{Tr}|U|$. The experiment involves a single trapped ion and two motional modes. The conceptual idea is that U is performed on a second system controlled by Bob and Alice is allowed send and receive quantum information to and from Bob's system. The authors make an analogy to an API. The paper will be of interest to Nature Communications readers."

"The experiment is interesting and includes a number of gate compilation methods to simplify the procedure. The authors clearly explain in the paper and the supplementary materials how the experiment is performed. The data is well presented and the error bars are explained. The error with decoherence is quite large and points to the limits of this experiment."

We are delighted the referee found them paper sufficiently interesting for readers of Nature communications. Indeed, we agree with the referee that the errors for decoherence is significant. In methods, we have included a more detailed discussion of how the effects of these errors can be mitigated by the client by calibrating her experiment, and that this process results in only a polynomial overhead in computational costs. We hope that this extra addition will help illustrate that though our experiment suffers significant decoherence, the adverse effects of decoherence can be efficiently overcome.

"The weakness of the paper is the API motivation. It seems unnecessary and then the readers expectations are deflated by the procedure occurring in a single system. The authors also claim in the discussion that the modes of motion provide a favorable path to scaling over the internal degrees of freedom but this is poorly motivated."

We thank the referee for this astute criticism. Indeed, on rereading the manuscript, we realized that we indeed over-emphasized distributed quantum computation in our motivation unnecessarily, causing a deflation in reader expectation when our experiment is performed on a single system. In response, we have retooled our introduction to provide a more balanced motivation, such that it is more readily apparent that off-loading part of a computation to a third party is useful even when only a single quantum system is involved. We believe this change has significantly helped align reader expectations. We sincerely thank the referee for helping instigate the improvement. We also agree that our previous discussion on favorable scaling could be improved and have now included a far more detailed discussion on scaling in the newly revised supplementary materials.

"In the supplementary material, Figure 2d has a typo in the first column of the fourth row. Both states in the bottom register should be 1."

Very astute! In the revision, the Fig. 2 is changed to Fig. 1 and the error has been corrected.

清华大学交叉信息研究院

Tsinghua University Institute for Interdisciplinary Information Sciences

北京市海淀区清华大学 FIT 楼, 邮编 100084, 电话: (010)62781693, 传真: (010)62797331

Tel: +86-10- 62781693

Fax: +86-10-62797331

Website: <http://iis.tsinghua.edu.cn>

Reply to Reviewer #2

"In the manuscript "Modular Quantum Computation in a Trapped Ion System" the authors describe an experimental demonstration of the modular quantum computation protocol detailed in [15] using a trapped Yb ion. The procedure is a variant of the DQC1-procedure described in [2] which allows a client to evaluate the modulus of the trace of a unitary. The unitary is provided as a service by a server and applied to a qubit provided by the client via a quantum-API. The modularity stems from the fact that the client can be blind to the particulars of the implementation of the unitary provided by the server. This is a highly relevant topic in quantum computing since all current realizations show only classical modularity, i.e. different circuit elements can be rearranged, or ion-qubits moved."

"The paper is well written and clearly structured, with very neat illustrations. Some of the implementation only becomes understandable when studying the supplementary material. I think the result is highly relevant to the field of quantum information processing with trapped ions and of interest to anyone working in implementations of quantum algorithms and quantum computing hardware and architecture. I am not convinced it warrants the interest of the general natural science community for the following reason."

"To me, the most impressive achievement is that the authors perform what is essentially a 3-qubit algorithm in a single-ion experiment by clever use of the different degrees of freedom provided by the system, namely the two qubit states, plus two auxiliary electronic Zeeman states, and 4 different motional states in the two radial modes. This shows how efficiently quantum information can be encoded in a trapped ion system. It also nicely illustrates the idea that in a server-client relationship, the implementation of a subroutine might be executed on a different kind of qubit platform with the exchange being handled by the API."

"This approach also appears to me to contain a weak point of the paper. Since there is only one real qubit, i.e. quantum system with complete gate set, the unitary has to be executed on that one qubit. So the information Alice holds on her control qubit is stored temporarily elsewhere, and the qubit the server receives is copied back onto Alice's qubit, so that the same control field (uWave) that she used to prepare her state performs the unitary. This seems to me to go against the spirit of modularity the authors themselves have evoked since the operations of the server are not independent and arbitrary, they are in fact the same as the client's. This removes the main complication from a quantum API, namely how to achieve the interface between the qubit-sender and the qubit-receiver to ensure continuous coherent control. Since the two controls are identical here, the problem reduces to that of a regular algorithm with operations performed in a single module. While it makes the implementation work (which is, I reiterate, very impressive), I worry that it also weakens the point the authors are trying to make with the demonstration - modularity (i.e. actual modularity rather than modularity in principle)."

We thank the referee for the feedback. We are most happy to hear that the referee found our experiment to be impressive. Indeed we were also excited to be able to perform a 3-qubit experiment on a single trapped ion.

清华大学交叉信息研究院

Tsinghua University Institute for Interdisciplinary Information Sciences

北京市海淀区清华大学 FIT 楼，邮编 100084，电话：(010)62781693，传真：(010)62797331

Tel: +86-10- 62781693

Fax: +86-10-62797331

Website: <http://iis.tsinghua.edu.cn>

The referee’s primary concern then is that he felt – while impressive – the choice of implementation appears not to mesh well with the spirit of modularity. In particular, while we were able to encode 3 qubits in the same ion, only a single 2-level degree of freedom (namely the internal state of the ion) had a complete set of gates. Consequently, our simulated ‘server’ during the benchmarking phase made use of the same Hilbert space as that of our client.

We sympathize with the referee’s perspective, and agree that under ideal settings, our experiment should involve multiple ions such that information is clearly sent from client to server. However, we would like to clarify the experiment nevertheless still does align with the spirit of modularity. Admittedly, it is our fault for not being sufficiently clear. In particular, on rereading our previous manuscript, we realized that it was not sufficiently comprehensive about how modular computation is defined. We only motivated modularity from the perspective of a client and remote server without emphasizing the importance of the concept in single systems and glossed over how the additional SWAP gates introduced in the experimental can fit in with our theoretical description of modular DQC1.

Let us explain first explain these details here, and then mention how we addressed these shortcomings in the revised manuscript.

We begin with a slightly more formal treatment of modular computation – a core motivation being that we do not need to reinvent the wheel. When we build a complex device, say a computer for example, we generally make it by connecting many prefabricated components (e.g. Hard Drive, RAM). Different manufactures make these devices, and we can mix n’ match them; as they communicate to each other via pre-defined interfaces. In a similar vein, when we build a complex algorithm, we can make use of many pre-designed functions as subroutines (e.g. inverting a linear equation) by knowing their API without needing to know the specifics of how these algorithms are designed. Of course, the client – server scenario mentioned in the manuscript is one such application, allowing Alice to offload part of a complex computation remotely to a third party (Bob) without knowing the specifications of how Bob chooses to implement this computation.

In the context of quantum computing, modular algorithms can be formalized as follows. We can formally describe an algorithm on the side of the client (Alice) as a sequence of elementary operations, where each operation could either be

1. A basic quantum operation Alice performs on her end (e.g. an elementary gate, some single-qubit measurement)
2. Calling on the server to implement U on the Hilbert space designated by the pre-agreed interface.

Figure 1: Circuit Representation of a Modular Computation (from Ref. [15])

清华大学交叉信息研究院

Tsinghua University Institute for Interdisciplinary Information Sciences

北京市海淀区清华大学 FIT 楼，邮编 100084，电话：(010)62781693，传真：(010)62797331

Tel: +86-10- 62781693

Fax: +86-10-62797331

Website: <http://iiis.tsinghua.edu.cn>

The goal of building a modular algorithm is then to find a sequence of such elementary operations to realize some U independent quantum device that implements P_U for whichever unitary U the server chooses to realize. In the circuit picture, we can think about this in quantum circuit picture as a quantum comb (see Figure 1). Alice's device is essentially a super-operator that takes as input a unitary U , and outputs a quantum channel $P[U]$. The key here is that Alice's sequence of operations itself is U independent. When this is satisfied, Alice's device enables the following properties (summarizing from Ref [15])

- **Independence of realization:** If Alice finds a new server, Charlie, which computes U using a more efficient algorithm, Alice's device can make use of Charlie's service without any modification—provided Charlie adheres to the same public interface.
- **Independence of function:** If Alice wishes to implement P_V instead of P_U , she does not need to modify her device. Alice needs only find another server that chooses to realize V instead of U on the other side of the interface.

From a more physical perspective, Alice can treat Bob's device as a black-box. She does not need to know anything about its internal structure or implementation.

In the introduction of our prior manuscript, we motivated this modular architecture almost exclusively from the perspective that Alice is a client, and Bob is a remote server. Indeed, we see this as one a primary application of the paradigm. As aforementioned though, modular architectures are also important in many other settings. Examples include

- *In metrology settings*, the unitary U could represent some unknown unitary process (e.g. some unknown magnetic field) that we are trying to probe. That is, Bob, is essentially a natural process. In this case, we are essentially computing something that is a function of U , while treating U as a black-box. The iconic case for example, being when $U = e^{iH\theta}$ and we wish to find information about either θ , or H , or both.
- *In programming Languages*, supplied software libraries contain pre-designed functions. For example, it is common to load packages available from a software library at the beginning of your program, which give you access to preprogramed functions which can be invoked throughout your code (for instance in a for loop). Here modularity allows Alice to call upon these pre-built functions as subroutines to design more complex algorithms, while treating these functions themselves completely as a black box. If the software library updates (presumably because they found a more efficient means of implementing a function), Alice would not need to modify her own code. In this case, Bob is simply a software library that runs locally on Alice's computer.

Notably, in these settings, Bob does not necessarily implement U on a different system from that of Alice. The key property of modular architectures – and what we would classify as the 'spirit of modularity' – is that Alice's device is *the same* irrespective of

- (a) what unitary U is being implemented,
- (b) how Bob chooses to implement this U .

The only thing Alice and Bob need to agree on is how to communicate quantum information between them (as specified by the interface). Operationally, this means that Alice's actions in pre-processing and post-processing do not change depending on (a) and (b). For these reasons, Alice can remain oblivious about (a) and (b).

On why our experiment satisfies the spirit of modularity – It is true that our experiment deviates from the theoretical circuit outlined in Figure 1 of the manuscript. However, we could recast our experimental circuit as follows

As can be seen, Alice's procedure for pre-processing and post-processing will be identical regardless of which unitary U the server chooses to implement, or how the server decides to synthesize U (e.g. what pulse sequences) – as long as they pre-agree of the interface. As such, we believe our experiment remains true to the spirit of modularity. We hope that with this information the referee can see our perspective.

Manuscript Changes – We revised the manuscript to convey the above concepts clearly. In particular, it makes far more sense to include the swap gates between the control and motional qubits as actions done by the client, rather than actions done by the server. In response we have

- Included a more balanced motivation of modular computation, so that it is more apparent that modular computation is important even in non-distributed scenarios.

清华大学交叉信息研究院

Tsinghua University Institute for Interdisciplinary Information Sciences

北京市海淀区清华大学 FIT 楼, 邮编 100084, 电话: (010)62781693, 传真: (010)62797331

Tel: +86-10- 62781693

Fax: +86-10-62797331

Website: <http://iis.tsinghua.edu.cn>

-
- Introduced a new methods section, whereby we review the definition of modularity in more detail to avoid ambiguity regarding its formal meaning.
 - Included a more detailed description of our experimental implementation – so that it is clearer how the set-up fits within the framework of modular computation.
 - Revised supplementary materials to emphasize the independence of Alice’s actions on the actions of Bob.

We include an additional marked-up pdf version of the manuscript to show such modifications.

"I have enjoyed reading this paper and believe it demonstrates an interesting quantum algorithm that touches on a subject not previously experimentally investigated, but for the reason stated, a more quantum computer focused journal like Nature-QI might be a better venue for this work."

We are delighted the referee has enjoyed our manuscript. It is indeed a subject that has not being previously investigated. While we agree that the previous manuscript would certainly suit the readers of npj:QI, we feel that it would also be interesting to a broader readership for the following reasons

1. While we absolutely agree that our experiment could have room for improvement (such as putting server and client on spatially separated systems), we do believe that our experiment does demonstrate the key properties behind modular architectures as explained above.
2. We think that modular architectures are important beyond just the server-client framework and have an impact in other areas of quantum science (e.g. metrology as aforementioned).

As such, we hope the referee can also see that potential suitability of our work for the audiences of Nature Communications.

Smaller points:

"- p.2 "The physical implementation involves pioneering techniques to coherently swap two motional modes of an ion trapped in a 3D harmonic oscillator, controlled on the internal levels of the trapped ion." - While it is true that this particular implementation of a C-Swap gate is new, storing information temporarily in the motional modes to create quantum gates is an old idea, as this very process lies at the heart of the Cirac-Zoller gate. I feel "pioneering" is too strong a term."

We thank the referee for pointing this out. Our previous statement was indeed only meant to emphasize the novelty of how we implemented the C-SWAP gate and had intention of implying that we developed the idea of creating quantum gates for storing information temporarily in motional modes. In response we have changed the sentence to.

"The physical implementation involves a new implementation of the C-SWAP gate, coherently swapping two

清华大学交叉信息研究院

Tsinghua University Institute for Interdisciplinary Information Sciences

北京市海淀区清华大学 FIT 楼，邮编 100084，电话：(010)62781693，传真：(010)62797331

Tel: +86-10- 62781693 Fax: +86-10-62797331 Website: <http://iis.tsinghua.edu.cn>

motional modes of an ion trapped in a 3D harmonic oscillator, controlled on the internal levels of the trapped ion."

"- p.2 "(e.g. coherence in the photon number degree of freedom)" - at this point in the paper, the experimental system has not been introduced, which photon degrees of freedom are meant is not clear"

The referee has a point. The sentences in parenthesis are now removed.

"- supplementary materials sections are labelled differently than the references to them in the manuscript (i.e. where is Supplementary II.C, referenced on p.3 bottom left?)"

Thanks, this has been fixed.

"- figure 5: why are the experimental errors making all trace estimates systematically lower, rather than e.g. pushing them to 0.5 or some other value?"

This is a good question, and upon review, it is indeed true that this was not addressed with sufficient clarity. The reason that the error pushes everything uniformly downwards is because the dominant source of the error is loss of coherence in the control qubit during Alice's use of the C-SWAP operation and the final measurement of the control qubit is in the X-basis. More precisely, the theoretical state (when expressed as density matrix with the first qubit as a control) at the time of Alice's measurement should ideally be

$$\rho_{ideal} = \frac{1}{2^{2n+1}} \begin{bmatrix} I^{\otimes 2n} & U \otimes U^\dagger \\ U^\dagger \otimes U & I^{\otimes 2n} \end{bmatrix},$$

whereby measuring the control qubit in the X-basis would have an expectation value $\langle \sigma_1 \rangle_{ideal} = |T(U)|^2$

However due to fluctuations in laser intensity and polarization, together with that of the magnitude field and trap frequency – coherence is lost on the control qubit. This results in a dampening of the off-diagonal terms. Therefore, in our experiment, the output state takes the form

$$\rho_{exp} = \frac{1}{2^{2n+1}} \begin{bmatrix} I^{\otimes 2n} & (1-\epsilon)U \otimes U^\dagger \\ (1-\epsilon)U^\dagger \otimes U & I^{\otimes 2n} \end{bmatrix} = (1-\epsilon)\rho_{ideal} + \epsilon \frac{I}{2^{2n+1}},$$

where $\epsilon > 0$ is the error rate that ranges between 0 and 1 and I is the identity matrix. We then see that that

$$\langle \sigma_1 \rangle_{exp} = (1-\epsilon)\langle \sigma_1 \rangle_{ideal} = (1-\epsilon)|T(U)|^2$$

Therefore, when we estimate $|T(U)|^2$ in experiment by the expectation value of $\langle \sigma_1 \rangle_{exp}$, our conclusion will also be scaled by a factor of $\lambda = 1 - \epsilon$, which is always lower than the true value of $|T(U)|^2$.

This is why it is possible to have get an asymptotically accurate estimate of $|T(U)|^2$ simply by setting

$$T(U)_{est} = \frac{\langle \sigma_1 \rangle_{exp}}{\lambda},$$

with a polynomial overhead in λ .

The referee's comments made us realize that our manuscript would benefit from a more careful discussion of these details. Therefore the revised manuscript features an addition section in methods describing these errors, and how Alice can account for them.

"- fig 5 b and accompanying text: Rescaling of the experimental values works here because the error of implementing a single-qubit unitary is small and the errors come from the operations associated with the pre- and postprocessing operations. I imagine it will be much harder when the server uses an unknown number of gates to make some unknown U . Does this not mean the correction will become server- and server-implementation dependent which seems to go against the API philosophy (i.e. "swapping one server for another") that the paper is based on?"

We thank the referee for bringing up this point. Indeed, the correction in our experiment is possible because its dominant source comes from the client, Alice. If the error came from the server, and dependent on which U the server implements, it would be much harder for Alice to account for.

We note however, that this sort of server dependence does not go against the API philosophy. Consider first a classical example. Suppose we are to design a program to evaluate the von Neumann entropy that calls the API to evaluate the eigenvalue of a matrix. Our program is considered to be correct as long as it evaluates the entropy whenever the server that our API connects, gives us correct eigenvalues. In particular

- The core philosophy of having an API here is that our program works correctly, *on the proviso* that the server delivers its part of the promise (e.g. the correct eigenvalues).
- Even if our program is flawless, it would still give incorrect outputs if the server returned erroneous eigenvalues.

In fact, if our program was able to correct the server's internal operational errors, the no-go theorem in Ref. [15] would state that making our program modular would be impossible. The intuition being that to correct for the server's errors, our program would need to know something about the server's implementation – and thus be tailor made for that server. This would necessarily make it incompatible with other alternate servers fielding different implementations (and thus prevent the ease of swapping servers).

It is same in the quantum regime. To adhere to the philosophy of an API, Alice would need to construct a U -independent quantum device that correctly estimates $|T(U)|$, by outsourcing the implementation of U to a third

清华大学交叉信息研究院

Tsinghua University Institute for Interdisciplinary Information Sciences

北京市海淀区清华大学 FIT 楼, 邮编 100084, 电话: (010)62781693, 传真: (010)62797331

Tel: +86-10- 62781693

Fax: +86-10-62797331

Website: <http://iiis.tsinghua.edu.cn>

party via the API.

The core philosophy here is that Alice's device can operate correctly (i.e. give an accurate estimate of $|T(U)|$), *on the proviso* that the server delivers U correctly via the pre-agreed API. In our experiment, we demonstrated that our device was able to account for its own errors through rescaling, using a sequence of elementary actions that remains the same regardless of U . As such this error mitigation procedure does adhere to the philosophy of APIs.

In light of the referee's comments, we realized that these ideas have not been made sufficiently transparent in the original manuscript. We hope the new subsection on error mitigation under 'Methods' will help rectify this shortcoming.

"- in the supplementary, the preparation of the mixed state is not completely clear to me: It is created by a running different circuits, one for each of the 4 basis states and averaging the results. Are 1000 repetitions performed for each circuit and the result averaged, or 250 for each giving 1000 in total? Also, if this is part of a larger quantum program with multiple API calls, running it in sequence for each combination might not be practical. Is there a way to create a mixed state resource online (perhaps by using another motional mode that is traced out?)"

We thank the referee for pointing this out and agree that the relevant section of the supplementary could be expressed more clearly. Firstly, the experiment was performed with 1000 repetitions for each circuit, and thus 4000 in total.

On the question of scaling, our present method of preprocessing can in fact scale efficiently in principle (of course, there would still be many practical challenges). This may come at first surprising. In particular, to evaluate $|T(U)|$ for a $2^n \times 2^n$ matrix, we are dealing with a $2n + 1$ qubit circuit, where there are 2^{2n} different combinations. Thus, fabricating these combinations would certainly be infeasible for large n .

However, in practice our design does allow an efficient means to circumvent this problem. In particular, let the circuits for each of these 2^{2n} combinations be labeled by two n -digital binary strings, l and m and denoted $C_{l,m}$. It is possible to show that the following protocol generates the required pre-processing state can be synthesis as follows

1. Generate two n -bit string uniformly at random, call these l and m
2. Synthesis $C_{l,m}$, where each $C_{l,m}$ itself can be built up by rearranging 4 very basic building blocks: The X gate, the H gate, and two specific 3-qubit composite gates.

As such, if Alice repeats the above for each call to the server, and after K calls (where K scales as polynomial of n), we are then able to estimate $|T(U)|$ to some fixed accuracy. Note $T(U)$ is the average (mean) of the diagonal entries of U , and thus by the central limit theorem can be evaluated efficiently by sampling the

清华大学交叉信息研究院

Tsinghua University Institute for Interdisciplinary Information Sciences

北京市海淀区清华大学 FIT 楼，邮编 100084，电话：(010)62781693，传真：(010)62797331

Tel: +86-10- 62781693

Fax: +86-10-62797331

Website: <http://iiis.tsinghua.edu.cn>

diagonal entries of U . As $|T(U)|$ is simply the absolute value of the mean of the eigenvalues of U , this carries over to $|T(U)|$.

While this was briefly mentioned in the previous appendix, we realized that the exposition was certainly not clear. This has been remedied in the new version of accompanying supplementary materials – where we have included a new figure (Figure 2) that describes how preprocessing can be scaled efficiently.

"there are some language errors in the supplementary:

examples: p2: "for THE general n case," "requires the implementation of THE following unitary," p3: "requires A single sigma measurement", main text: figure 4 caption "is A consequence""

We thank the referee for pointing these out and we have corrected or rephrased them.

"figure 3: the bottom figure seems to be missing some colored circles, I assume the 3-cube sequence is meant to show the purple and red dot changing places while leaving the black and green dot unmoved by the end."

We may not have explained this sufficiently clearly. Here both the arrows and dots are meant to show the "processes." Dots are not populations or states. The arrow indicate that population is transferred, and the dots show the population is not transferred. We thank the referee for pointing out the ambiguity and now include the explanation in the caption.

stylistic comments (these are a matter of preference):

"The mention of companies Google and IBM in the first paragraph tries to emulate the introduction to [1] but doesn't do it well. The IBM quantum computer would perform worse on the circuit the authors have implemented here. Generic examples without commercial brands might be better."

We thank the referee for the suggestion. The mention of IBM has now been replaced with a generic example. However, we felt using Google as an example of remote services seemed to offer a particularly quick method to communicate the basic intuition of APIs – and have left this mention in.

"p.2 "no other promises are made." - too strong, a common phase reference is also necessary to make it work"

We thank the referee for pointing this out. Indeed, a common phase reference that Alice and Bob both agree on a designated computational basis is definitely required. This sentence aforementioned is now removed. Instead, we have included a newly created section in methods with a far more detailed definition of quantum interfaces. We believe this will help clarify all ambiguities.

Reviewers' Comments:

Reviewer #2:

Remarks to the Author:

The authors have made substantial changes to the manuscript in light of my comments, and have reformulated the motivation for the experiment and the scheme presented in figure 1. This makes the nature of the modularity they have in mind clear. The action of the server is independent of the context of Alice's circuit, and hence interchangeable in its particular implementation. This drops the emphasis on the server as a different/remote/etc. system, making it a black-box quantum subroutine. The reader can now make those connections from the manuscript, and I am of course still convinced of the merit of the experimental realization.

I recommend publication in Nature Communications.

typo in fig. 1 caption: "initialized in state $|0\rangle$ " should be $|+\rangle$